✦ PLOS | ONE

# Precision medicine and actionable alterations in lung cancer: A single institution experience

**Isa Mambetsariev[1], Yingyu Wang[2], Chen Chen[2], Sorena Nadaf[2], Rebecca Pharaon[1], Jeremy Fricke[1], Idoroenyi Amanam[1], Arya Amini[3], Andrea Bild[1], Peiguo Chu[4], Loretta Erhunmwunsee[5], Jae Kim[5], Janet Munu[2], Raju Pillai[4], Dan Raz[5], Sagus Sampath[3], Lalit Vora[6], Fang Qiu[7], Lynette Smith[7], Surinder K. Batra[8], Erminia Massarelli[1], Marianna Koczywas[1], Karen Reckamp[1], Ravi Salgia[1] ***

**1** Department of Medical Oncology and Therapeutics Research, City of Hope, Duarte, California, United States of America, **2** Center for Informatics, City of Hope, Duarte, California, United States of America, **3** Department of Radiation Oncology, City of Hope, Duarte, California, United States of America, **4** Department of Pathology, City of Hope, Duarte, California, United States of America, **5** Department of Thoracic Surgery, City of Hope, Duarte, California, United States of America, **6** Department of Diagnostic Radiology, City of Hope, Duarte, California, United States of America, **7** Department of Biostatistics, University of Nebraska Medical Center, Omaha, Nebraska, United States of America, **8** Department of Biochemistry and Molecular Biology, University of Nebraska Medical Center, Omaha, Nebraska, United States of America

* rsalgia@coh.org

**Data Availability Statement:** All relevant study data are within the manuscript and its Supporting Information files. Any additional data is available

## Abstract

### Objectives

Oncology has become more reliant on new testing methods and a greater use of electronic medical records, which provide a plethora of information available to physicians and researchers. However, to take advantage of vital clinical and research data for precision medicine, we must initially make an effort to create an infrastructure for the collection, storage, and utilization of this information with uniquely designed disease-specific registries that could support the collection of a large number of patients.

### Materials and methods

In this study, we perform an in-depth analysis of a series of lung adenocarcinoma patients (n = 415) with genomic and clinical data in a recently created thoracic patient registry.

### Results

Of the 415 patients with lung adenocarcinoma, 59% (n = 245) were female; the median age was 64 (range, 22–92) years with a median OS of 33.29 months (95% CI, 29.77–39.48). The most common actionable alterations were identified in EGFR (n = 177/415 [42.7%]), ALK (n = 28/377 [7.4%]), and BRAF V600E (n = 7/288 [2.4%]). There was also a discernible difference in survival for 222 patients, who had an actionable alteration, with a median OS of 39.8 months as compared to 193 wild-type patients with a median OS of 26.0 months (P<0.001). We identified an unprecedented number of actionable alterations [53.5% (222/415)], including distinct individual alteration rates, as compared with 15.0% and 22.3% in TCGA and GENIE respectively.

through TCGA (https://www.cancer.gov/tcga) and AACR GENIE (https://www.aacr.org/professionals/research/aacr-project-genie/aacr-project-genie-data/).

**Funding:** Research reported in this publication included work performed in the Core facilities, Biostatistics Core and Center for Informatics Core, supported by the National Cancer Institute of the National Institutes of Health under award number P30CA033572. Pilot Projects and Early Phase Clinical Research Support reported in this publication was supported by the National Cancer Institute of the National Institutes of Health under award numbers R01CA218545 and U54CA209978. The content is solely the responsibility of the authors and does not necessarily represent the official views of the National Institutes of Health. The American Association for Cancer Research provided its financial and material support in the development of the AACR Project GENIE registry. The funders had no role in study design, data collection and analysis, decision to publish, or preparation of the manuscript.

**Competing interests:** The authors have declared that no competing interests exist.

**Abbreviations:** NSCLC, Non-small cell lung cancer; IRB, Institutional Review Board; FISH, Fluorescent in situ hybridization; LUAD, Lung adenocarcinoma; NGS, Next-Generation Sequencing; TKIs, tyrosine-kinase inhibitors; FDA, U.S. Food and Drug Administration; FFPE, formalin-fixed paraffin-embedded; IHC, immunohistochemistry; PCR, polymerase chain reaction; PFS, progression-free survival; THOR, Thoracic Oncology Registry; OS, overall survival; SWI/SNF, SWItch/Sucrose Non-Fermentable; NCCN, National Comprehensive Cancer Network; ASCO, American Society of Clinical Oncology.

## Conclusion

The use of patient registries, focused genomic panels and the appropriate use of clinical guidelines in community and academic settings may influence cohort selection for clinical trials and improve survival outcomes.

## Introduction

In response to advances in genomic testing and the rapid integration of new drugs and publications, oncologists have been adapting the concept of precision medicine where evidence-based medicine guides treatment decisions for individuals [1]. However, more effort is required to translate these benefits into real-world durable survivals for patients. Therefore, in this pursuit, several organizations have implemented the utilization of guidelines and pathways to ensure that patients receive proper testing and are assigned to proper treatment options, which in theory should then translate into durable survival outcomes [2–6]. This spur towards personalized medicine is primarily driven by the advances in genomic testing, biomarker-driven therapy, and immunotherapy that have transformed the landscape of oncology care and have greatly improved outcomes for patients [7–12]. Next-generation sequencing (NGS) has been highly influential in its ability to identify genomic alterations that confer sensitivity to approved and investigational targeted therapies in patients suffering from a variety of advanced stage cancers. The application of molecular testing is transforming cancer into a diverse template of genomic alterations that drive oncogenesis [13].

In view of the vast clinical data offered by NGS in non-small cell lung cancer (NSCLC), City of Hope (COH) has established a concise and efficient patient registry (Registry of Hope) for the collection of genomic alterations and outcomes-focused clinical data. Here we present the results of genomic testing performed as part of routine clinical care and correlative analysis of 415 lung adenocarcinoma (LUAD) patients within the Thoracic Oncology Registry (THOR). We hypothesized that the application of broad genomic testing provides not only a comprehensive overview of clinical heterogeneity in lung cancer but may also guide the future of oncology care as more and more precision medicine therapeutics emerge. In this study, we also evaluate the genomic profile of our COH cohort as it compares to national testing results found in GENIE/TCGA databases.

## Materials and methods

### Patients

Patients with advanced LUAD (n = 415), were enrolled in this analysis and evaluated at COH from 2008 to 2016, the data was collected between 2016 and 2018 through retrospective chart review on patients who had LUAD diagnosis and molecular testing performed at the discretion of their primary clinical provider. All 415 patients had metastatic disease, with 89 percent of patients who presented with metastatic disease at the time of initial diagnosis while others later developed metastatic disease. THOR was used to perform this study and data was collected into the registry over time on eligible patients. Different NGS platforms were used as described in S1 Table. Patients were categorized by race/ethnicity according to what they had reported to their oncologist and the data was pulled from the electronic medical record. The categories included African American, American Indian or Alaska Native, Asian, White, Native Hawaiian or Other Pacific Island, Other, and Unknown/Declined to answer. Confounding variables,

such as socioeconomic status, were not adjusted but the researchers felt that it was important to understand the diverse demographic makeup of the lung cancer patients and how it relates to their mutational status. The study was approved by the City of Hope institutional review board and in accord with an assurance filed with and approved by the Department of Health and Human Services at COH. This study was approved by the Institutional Review Board at COH under IRB 18217 and was conducted according to the Declaration of Helsinki. Data was de-identified and analyzed anonymously.

## Genomic analysis

Patient genomic alteration data was collected manually from clinical genomic tests alongside the clinical information. The molecular testing results in this study were all performed at around the time of metastatic diagnosis and did not include multiple time points. Clinical actionability of genomic alterations was assessed for each patient and included genomic alterations that had US Food and Drug Administration (FDA) approved targeted therapy in NSCLC. The actionable genomic alterations in this study were defined as EGFR exon 19 deletions and L858R mutation, ALK rearrangement, ROS1 rearrangement, NTRK fusions, and BRAF V600E alterations based on FDA-approved therapy available. MET exon 14 splice site/ deletion was also included as actionable based on FDA accelerated path to approval [14, 15]. These alterations were chosen because they are largely exclusive of one another. Any patient who had at least one of these six gene alterations was considered to have an actionable alteration and anyone whose genomic testing results did not identify any of the six gene alterations was considered wild-type. When individual genes were evaluated for survival and various statistical analyses the patients with alterations were only compared with patients who were tested negative for that specific gene. Tile plot maps were generated using *seaborn* library for Python (version 2.7.14) [16].

## Comparison with TCGA and GENIE

For the alteration rate of the THOR data set, each gene that had an alteration was calculated as an individual patient divided by the number of patients who were tested for that gene. Genomic alteration data were retrieved from TCGA (study id = luad_tcga_pub) and GENIE (GENIE Cohort v4.1-public) [17, 18]. For each gene, only samples profiled in all molecular profiles including copy-number alteration, somatic mutations, and structural rearrangement were counted. Alteration rate for each gene was calculated by number of samples with at least one alteration divided by the number of profiled samples.

## Statistical analysis

Patient characteristic parameters were evaluated using the $\chi 2$ test to test for association between characteristic values and age. OS was calculated from the date of diagnosis of metastatic disease to death or last follow-up visit. Patients who were thought to be alive at the end of the study were censored at the time of the last visit. Survival estimates for the studied patients were generated using the Kaplan-Meier method and the Cox proportional hazards models were used to estimate the hazard ratios. The analysis was performed using *survival* and *survminer* packages for R software (version 3.4.4) and SAS version 9.4 (SAS Institute, Cary, NC, United States). The assumption of proportional hazards was tested using goodness-of-fit tests, graphical methods and time-dependent variable methods. The extended Cox models using time-dependent variables were used to adjust the non-proportionality of variables. By adding interaction terms between time and the variables that violated the proportional hazards assumption, the models allow for the possible diverging survival curves over time.

## Results

### Patients

415 eligible patients with LUAD and tumor genotyping results were identified in THOR at the time of this study. The majority of patients were Stage IV (n = 369/415 [89%]) at the time of initial diagnosis and 46/415 (11%) patients were stage I-III at diagnosis who eventually recurred as metastatic disease. There were 281 patients who underwent broad-based genomic testing (more than 30 genes) using NGS and 134 patients were tested for a few genes in a small panel (less than 10 genes). 254 (61%) patients were never-smokers (n = 212) or had a smoking history of fewer than 10 pack-years (light smokers; n = 42) and 161 (39%) patients were smokers with a history of 10–29 pack-years (medium smokers; n = 74) or > = 30 pack-years (heavy smokers; n = 87). The overall median age at diagnosis was 64 (22–92) and the median OS was 33.29 months (95% CI, 29.77–39.48), with the majority of patients being female (n = 245 [59%]). Detailed patient characteristics noted in Table 1.

### Genomic alterations

There were 323 different genes with evidence of genomic alteration in this group of patients. The most commonly occurring alterations in oncogenes were found in EGFR (n = 207/415 [50%]), KRAS (n = 97/352 [28%]), and ALK rearrangement (n = 28/377 [7%]), while the most commonly occurring tumor suppressor genes consisted of TP53 (n = 140/283 [49%]), LRP1B (n = 63/228 [28%]), and STK11 (n = 39/278 [14%]) (Fig 1A). The median number of genes altered in patients who underwent broad-based sequencing was 10. The most common actionable alterations were identified in EGFR L858R/exon 19 deletion (n = 177/415 [42.7%]), ALK rearrangement (n = 28/377 [7.4%]), ROS1 rearrangement (n = 3/257 [1.2%]), BRAF V600E (n = 7/288 [2.4%]), and MET exon 14 splice site/deletion (n = 7/287 [2.4%]). No NTRK fusion alterations were identified in our cohort. 86% (177/207) of patients with EGFR alterations had an actionable alteration while 14% (29/207) consisted of exon 18 mutations, exon 20 insertions and other EGFR alterations including amplifications or substitution variants of unknown significance. EGFR alterations were split into five subtypes with the majority of Asians (n = 51/99 [52%]) and whites (n = 46/95 [48%]) presenting with an exon 19 deletion. KRAS alterations predominated in whites (n = 77/97 [79%]) and patients with history of medium (n = 24/97 [25%]) or heavy smoking (n = 47/97 [48%]) (Table 2).

### Comparison with TCGA and GENIE

While the gene alteration rates of LUAD patients from TCGA/GENIE were comparable [17, 18], evident differences were observed between the patients of THOR and these two public consortiums (Fig 1B). Importantly, a large proportion of patients had actionable alterations [53.5% (222/415)] with the highest rate of actionable EGFR mutations at 42.7% (177/415), as compared with 9.1% and 17.1% actionable rate in TCGA/GENIE respectively. In particular, our dataset had a significantly increased prevalence of EGFR (49.9%), BRCA2 (14.1%), and chromatin modifying genes (ARID1B [18.8%], ARID1A [17.7%], MLL2 [13.6%] and MLL [13.2%]) as compared to the other datasets (Table 3). Analysis of subgroups of patients in GENIE that were diagnosed with or without metastasis showed similar gene alteration rates of these genes, indicating the difference we observed between THOR and TCGA/GENIE patients was not directly linked to metastasis (Table 3).

### Survival

The median OS of all patients was 33.3 months (95% CI; 29.8–39.5) and female patients had a better median OS of 39.8 months (95% CI, 33.5–45.5 months; HR, 1.56; 95% CI, 1.21–2.00;

**Table 1. Patient characteristics.**

| Characteristic | Total, No. (%) | Age, No. (%) | | P Value | Chi-square Statistic |
|---|---|---|---|---|---|
| | | <70 | > = 70 | | |
| **Patients** | 415 (100) | 289 (100) | 126 (100) | NA | NA |
| **Sex** | | | | 0.341 | 0.906 |
| Female | 245 (59) | 175 (61) | 70 (56) | | |
| Male | 170 (41) | 114 (39) | 56 (44) | | |
| **Race** | | | | 0.127 | 9.956 |
| African American | 10 (2) | 6 (2) | 4 (3) | | |
| American Indian or Alaska Native | 2 (0) | 1 (0) | 1 (1) | | |
| Asian | 136 (33) | 107 (37) | 29 (23) | | |
| White | 247 (60) | 161 (56) | 86 (68) | | |
| Native Hawaiian or Other Pacific Islander | 3 (1) | 3 (1) | 0 (0) | | |
| Other | 12 (3) | 8 (3) | 4 (3) | | |
| Unknown/Declined to Answer | 5 (1) | 3 (1) | 2 (2) | | |
| **Ethnicity** | | | | 0.150 | 3.796 |
| Hispanic or Latino | 40 (10) | 28 (10) | 12 (10) | | |
| Not Hispanic or Latino | 369 (89) | 259 (90) | 110 (87) | | |
| Unknown/Declined to Answer | 6 (1) | 2 (1) | 4 (3) | | |
| **Smoking Status** | | | | 0.036 | 8.550 |
| Never | 212 (51) | 159 (55) | 53 (42) | | |
| Light (<10 pack years) | 42 (10) | 31 (11) | 11 (9) | | |
| Medium (10–29 pack years) | 74 (18) | 44 (15) | 30 (24) | | |
| Heavy (> = 30 pack years) | 87 (21) | 55 (19) | 32 (25) | | |
| **Stage** | | | | 0.004 | 13.595 |
| I | 15 (4) | 7 (2) | 8 (6) | | |
| II | 14 (3) | 5 (2) | 9 (7) | | |
| III | 17 (4) | 10 (3) | 7 (6) | | |
| IV | 369 (89) | 267 (92) | 102 (81) | | |
| **EGFR (L858R/exon 19 deletion)** * | | | | 0.059 | 3.559 |
| Alteration | 177 (43) | 132 (46) | 45 (36) | | |
| Tested Negative | 238 (57) | 157 (54) | 81 (64) | | |
| **ALK (Rearrangement)*** | | | | 0.001 | 10.349 |
| Positive | 28 (7) | 27 (10) | 1 (1) | | |
| Tested Negative | 349 (93) | 235 (90) | 114 (99) | | |
| **ROS1 (Rearrangement)*** | | | | 0.233 | 1.422 |
| Positive | 3 (1) | 3 (2) | 0 (0) | | |
| Tested Negative | 254 (99) | 172 (98) | 82 (100) | | |
| **BRAF (V600E)*** | | | | 0.319 | 0.995 |
| Positive | 7 (2) | 6 (3) | 1 (1) | | |
| Tested Negative | 281 (98) | 194 (97) | 90 (99) | | |
| **MET (exon 14 splice-site/deletion)*** | | | | 0.022 | 5.228 |
| Positive | 7 (2) | 2 (1) | 5 (5) | | |
| Tested Negative | 280 (98) | 194 (99) | 86 (95) | | |
| **KRAS** | | | | <0.001 | 12.410 |
| Positive | 97 (28) | 53 (22) | 44 (40) | | |
| Tested Negative | 255 (72) | 189 (78) | 66 (60) | | |

(*Continued*)

**Table 1.** (Continued)

| Characteristic | Total, No. (%) | Age, No. (%) | | P Value | Chi-square Statistic |
| --- | --- | --- | --- | --- | --- |
| | | <70 | >= 70 | | |
| **TP53** | | | | 0.306 | 1.046 |
| Positive | 140 (49) | 99 (52) | 41 (45) | | |
| Tested Negative | 143 (51) | 93 (48) | 50 (55) | | |

*Only patients who had genomic test results were counted for each gene. Total number of patients with ALK, ROS1, BRAF, MET, KRAS and TP53 tested were 377, 257, 288, 287, 352, and 283 respectively.

Table 4) as compared to median OS of 27.4 months for male patients. The OS was also better for never/light smokers (<10 pack years) with a median OS of 39.5 months (95% CI, 32.9–45.5 months; HR, 1.42; 95% CI, 1.10–1.83; Table 4) as compared to medium/heavy smokers (> = 10 pack years) who had a median OS of 25.7 months (95% CI, 22.1–33.3 months). Two genes had violations of the proportional hazards assumption EGFR in actionable and KRAS in actionable, therefore, we added an interaction with time to account for changes in the HR. When comparing the median OS of KRAS positive patient (25.5 months; 95% CI, 17.6–32.9 months) with the median OS of actionable patient who had tested KRAS negative (41.1 months; 95% CI, 39.3–55.0 months), the difference was noticeable (Fig 2A), the HR changes from high risk of death in the KRAS positive patients initially to less risk over time (Table 4). Of the six actionable genes only ALK rearrangement patients had a difference in survival with a median OS of 82.6 months (95% CI, 82.6-NR; HR, 0.35; 95% CI, 0.17–0.68;) as compared to 26.6 months (95% CI, 22.1–33.1 months) for tested ALK rearrangement negative patients (Fig 2B). The genomic testing panel size had no discernible OS difference and broad-based sequencing had a median OS of 33.4 months (95% CI, 29.2–44.2 months) as compared to 33.5

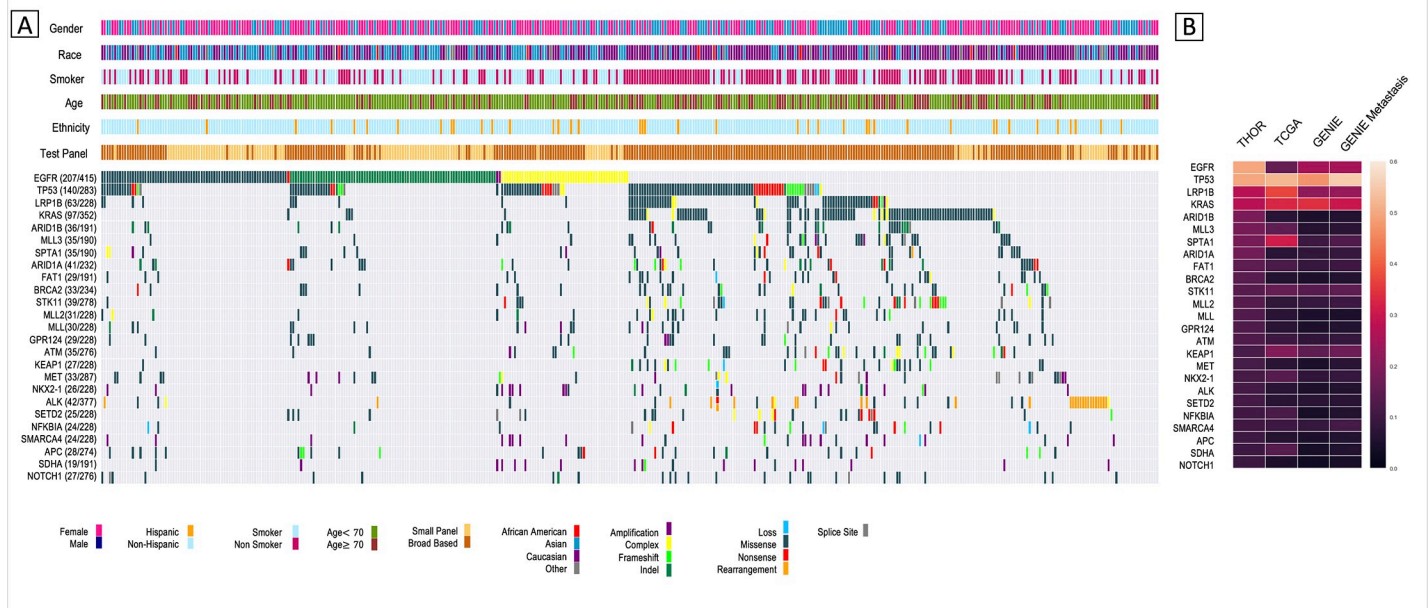

**Fig 1. Landscape overview of molecular profiling in 415 THOR lung adenocarcinoma patients.** A) Tile plot of top 25 most prevalent genes and their demographic parameters sorted by genomic alteration rate and subtypes, including amplification, complex (more than one type of alteration), frameshift, indel (insertion/deletion), loss, missense, nonsense, rearrangement, and splice site. For each gene, the alteration rate was calculated by number of patients with alteration divided by the number of patients tested for this gene. B) Heatmap showing a comparison of the genomic alteration rates between THOR, TCGA, GENIE, and GENIE Metastatic for genes in 1A.

**Table 2. Patients characteristic table for EGFR and KRAS alteration subtypes.**

| Characteristic | EGFR, No. (%)* | | | | | | KRAS, No. (%) | | | | |
|---|---|---|---|---|---|---|---|---|---|---|---|
| | Total | exon 18 mutation | exon 19 deletion | exon 20 insertion | exon 21 mutation | Others | Total | G12 | G13 | Q61 | Others |
| **Patients** | 207 (100) | 9 (4) | 106 (51) | 16 (8) | 72 (35) | 7 (3) | 97 (100) | 79 (81) | 7 (7) | 9 (9) | 2 (2) |
| **Sex** | | | | | | | | | | | |
| Female | 130 (63) | 4 (44) | 65 (61) | 12 (75) | 47 (65) | 3 (43) | 49 (51) | 39 (49) | 5 (71) | 4 (44) | 1 (50) |
| Male | 77 (37) | 5 (56) | 41 (39) | 4 (25) | 25 (35) | 4 (57) | 48 (49) | 40 (51) | 2 (29) | 5 (56) | 1 (50) |
| **Age** | | | | | | | | | | | |
| <70 | 155 (75) | 5 (56) | 84 (79) | 15 (94) | 48 (67) | 6 (86) | 53 (55) | 45 (57) | 3 (43) | 4 (44) | 1 (50) |
| > = 70 | 52 (25) | 4 (44) | 22 (21) | 1 (6) | 24 (33) | 1 (14) | 44 (45) | 34 (43) | 4 (57) | 5 (56) | 1 (50) |
| **Race** | | | | | | | | | | | |
| African American | 4 (2) | 0 (0) | 3 (3) | 0 (0) | 1 (1) | 0 (0) | 4 (4) | 3 (4) | 0 (0) | 1 (11) | 0 (0) |
| American Indian or Alaska Native | 0 (0) | 0 (0) | 0 (0) | 0 (0) | 0 (0) | 0 (0) | 2 (2) | 2 (3) | 0 (0) | 0 (0) | 0 (0) |
| Asian | 99 (48) | 6 (67) | 51 (48) | 6 (38) | 37 (51) | 2 (29) | 11 (11) | 9 (11) | 1 (14) | 0 (0) | 1 (50) |
| White | 95 (46) | 3 (33) | 46 (43) | 10 (63) | 32 (44) | 4 (57) | 77 (79) | 62 (78) | 6 (86) | 8 (89) | 1 (50) |
| Native Hawaiian or Other Pacific Islander | 0 (0) | 0 (0) | 0 (0) | 0 (0) | 0 (0) | 0 (0) | 1 (1) | 1 (1) | 0 (0) | 0 (0) | 0 (0) |
| Other | 8 (4) | 0 (0) | 6 (6) | 0 (0) | 1 (1) | 1 (14) | 1 (1) | 1 (1) | 0 (0) | 0 (0) | 0 (0) |
| Unknown/Declined to Answer | 1 (0) | 0 (0) | 0 (0) | 0 (0) | 1 (1) | 0 (0) | 1 (1) | 1 (1) | 0 (0) | 0 (0) | 0 (0) |
| **Ethnicity** | | | | | | | | | | | |
| Hispanic or Latino | 17 (8) | 0 (0) | 13 (12) | 1 (6) | 2 (3) | 1 (14) | 11 (11) | 10 (13) | 0 (0) | 1 (11) | 0 (0) |
| Not Hispanic or Latino | 188 (91) | 9 (100) | 92 (87) | 15 (94) | 69 (96) | 6 (86) | 84 (87) | 68 (86) | 6 (86) | 8 (89) | 2 (100) |
| Unknown/Declined to Answer | 2 (1) | 0 (0) | 1 (1) | 0 (0) | 1 (1) | 0 (0) | 2 (2) | 1 (1) | 1 (14) | 0 (0) | 0 (0) |
| **Smoking Status** | | | | | | | | | | | |
| Never | 137 (66) | 5 (56) | 72 (68) | 9 (56) | 50 (69) | 3 (43) | 17 (18) | 14 (18) | 1 (14) | 1 (11) | 1 (50) |
| Light (<10 pack years) | 23 (11) | 1 (11) | 11 (10) | 3 (19) | 7 (10) | 2 (29) | 9 (9) | 7 (9) | 1 (14) | 1 (11) | 0 (0) |
| Medium (10–29 pack years) | 39 (19) | 3 (33) | 21 (20) | 3 (19) | 11 (15) | 1 (14) | 24 (25) | 19 (24) | 1 (14) | 3 (33) | 1 (50) |
| Heavy (> = 30 pack years) | 8 (4) | 0 (0) | 2 (2) | 1 (6) | 4 (6) | 1 (14) | 47 (48) | 39 (49) | 4 (57) | 4 (44) | 0 (0) |

*Three patients carried two subtypes of EGFR alteration (e.g., exon 19 deletion and exon 18 mutation) and were counted two times.

months (95% CI, 27.5–41.2 months) for the small panel (Fig 2C). There was a discernible difference in survival for 222 patients, who had an actionable alteration (such as EGFR L858R/exon 19 deletion, ALK rearrangements, ROS1 rearrangements, BRAF V600E, NTRK fusions and MET exon 14 splice site/deletion), with a median OS of 39.8 months as compared to 193 patients who were wild-type with a median OS of 26.0 months (Fig 2D).

**Table 3. Genomic alteration rates of THOR referring to TCGA/GENIE/GENIE metastasis patients.**

| Gene | Altered | Tested | THOR Alteration Rate (n = 415) | TCGA Alternation Rate (n = 507) | GENIE Alternation Rate (n = 6529) | GENIE Metastasis Alternation Rate (n = 2697) |
|---|---|---|---|---|---|---|
| **EGFR** | **207** | **415** | **50%** | **16%** | **25%** | **25%** |
| TP53 | 140 | 283 | 50% | 52% | 47% | 55% |
| LRP1B | 63 | 228 | 28% | 37% | 22% | 23% |
| KRAS | 97 | 352 | 28% | 33% | 34% | 30% |
| **ARID1B** | **36** | **191** | **19%** | **7%** | **5%** | **6%** |
| MLL3 | 35 | 190 | 18% | 15% | 6% | 7% |
| SPTA1 | 35 | 190 | 18% | 31% | 10% | 13% |
| **ARID1A** | **41** | **232** | **18%** | **7%** | **8%** | **9%** |
| FAT1 | 29 | 191 | 15% | 12% | 9% | 10% |
| **BRCA2** | **33** | **234** | **14%** | **6%** | **5%** | **6%** |
| STK11 | 39 | 278 | 14% | 16% | 14% | 15% |
| **MLL2** | **31** | **228** | **14%** | **8%** | **9%** | **10%** |
| **MLL** | **30** | **228** | **13%** | **7%** | **5%** | **5%** |
| GPR124 | 29 | 228 | 13% | 7% | 5% | 6% |
| ATM | 35 | 276 | 13% | 10% | 8% | 9% |
| KEAP1 | 27 | 228 | 12% | 20% | 15% | 17% |
| MET | 33 | 287 | 12% | 7% | 5% | 7% |
| NKX2-1 | 26 | 228 | 11% | 14% | 8% | 9% |
| ALK | 42 | 377 | 11% | 7% | 5% | 6% |
| SETD2 | 25 | 228 | 11% | 7% | 7% | 7% |
| NFKBIA | 24 | 228 | 11% | 12% | 4% | 6% |
| SMARCA4 | 24 | 228 | 11% | 10% | 9% | 11% |
| APC | 28 | 274 | 10% | 6% | 5% | 6% |
| SDHA | 19 | 191 | 10% | 14% | 4% | 6% |
| NOTCH1 | 27 | 276 | 10% | 5% | 3% | 4% |

Genomic alteration data were collected for 415 THOR lung adenocarcinoma patients, 507 TCGA Lung Adenocarcinoma patients (study id = luad_tcga_pan_can_atlas_2018), 6529 GENIE Lung Adenocarcinoma patients (GENIE Cohort v5.0-public, cancer type detailed = lung adenocarcinoma), and 2697 GENIE metastasis lung adenocarcinoma patients (GENIE Cohort v5.0-public, cancer type detailed = lung adenocarcinoma, sample type = Metastasis).

## Discussion

The actualization of tyrosine kinase inhibitors, monoclonal antibodies, and immunotherapy drugs has pushed the treatment of lung cancer forward, however, the reality remains that lung cancer is the leading cause of cancer deaths [19]. Genomic testing has become more crucial in oncology care and recently there have been several studies launched that aim to match targeted therapy to patients based on their omic profile [20–27]. It is now a standard recommendation that patients with advanced NSCLC undergo routine molecular testing for identification of certain known genomic abnormalities, most notably ALK rearrangements, EGFR mutations, BRAF V600E, ROS1 rearrangements, and NTRK fusions [5]. More so several inhibitors for various markers, such as ERBB2, MET, and RET are quickly being implemented in standard clinical care and several others await FDA approval [28–32]. Thus, this study aims to highlight a unique single-site perspective into the clinical heterogeneity found in our cohort of patients that is largely composed of clinically relevant markers that will benefit from recent advances in precision medicine and a cohort that is distinctly different from other databases [17, 18]. To achieve durable survivals for patients it vital to implement appropriate testing at diagnosis and

**Table 4. Cox proportional hazard regression models for univariate survival.**

| Risk Factor | Median Survival (95% CI), months | Hazard Ratio (95% CI) | P Value |
|---|---|---|---|
| Sex, Male vs Female | 27.4 (20.6–31.7) vs 39.8 (33.5–45.5) | 1.56 (1.21–2.00) | <0.001 |
| Age, > = 70 vs <70 | 29.0 (19.5–36.5) vs 36.3 (31.4–42.9) | 1.41 (1.08–1.85) | 0.012 |
| Smoking Status, Medium + Heavy vs Never + Light | 25.7 (22.1–33.3) vs 39.5 (32.9–45.5) | 1.42 (1.10–1.83) | 0.006 |
| EGFR (Actionable)[a][b] | | | |
| Positive vs Negative | 39 (32.6–43.8) vs 26 (22.1–32.9) | 0.46 (0.31–0.68) | 0.0001 |
| EGFR x time | | 1.02 (1.01–1.03) | 0.0022 |
| ALK (Rearrangement)[b] | | | |
| Positive vs Negative | 82.6 (82.6-NR) vs 26.6 (22.1–33.1) | 0.35 (0.17–0.68) | 0.002 |
| ROS1 (Rearrangement)[b] | | | |
| Positive vs Negative | NR vs 26 (20.8–33.3) | - | - |
| BRAF (V600E)[b] | | | |
| Positive vs Negative | 73.4 (11.6-NR) vs 28.0 (22.2–34.4) | 0.70 (0.26–1.91) | 0.487 |
| MET (exon 14 Splice-site/Deletion)[b] | | | |
| Positive vs Negative | 17.2 (8.77-NR) vs 28.0 (22.2–34.4) | 0.59 (0.19–1.89) | 0.377 |
| KRAS in Actionable[c] | | | |
| Positive vs Negative | 25.5 (17.6–32.9) vs 41.4 (39.3–55.0) | 2.80 (1.70–4.61) | <0.0001 |
| KRAS x time | | 0.98 (0.97–0.99) | 0.0065 |
| KRAS in Non-Actionable[d] | | | |
| Positive vs Negative | 25.5 (17.6–32.9) vs 27.7 (22.2–42.9) | 1.06 (0.73–1.53) | 0.769 |
| TP53 in Actionable[c] | | | |
| Positive vs Negative | 25.5 (20.6–34.9) vs 67.6 (39.5-NR) | 1.87 (1.12–3.12) | 0.017 |
| TP53 in Non-Actionable[d] | | | |
| Positive vs Negative | 25.5 (20.6–34.9) vs 30.1 (21.6–54.0) | 1.13 (0.75–1.69) | 0.561 |

[a]. EGFR All patients had EGFR tested.

[b]. For each gene, patients carrying actionable alterations were compared with patients who were tested negative, with patients carrying actionable alterations from other genes excluded from its analysis.

[c]. For KRAS and TP53, patients who were tested positive with no actionable alteration were compared with patients who were tested negative but had actionable gene alterations.

[d]. For KRAS and TP53, patients who were tested positive were compared with patients who were tested negative and patients who had actionable gene alterations were excluded from both groups.

upon progression, and guided treatment plans (both guidelines and pathways) through consolidation of clinical data into patient registries [33].

## Precision genomics

In a recent study that compared the use of broad-based genomic sequencing versus small-panel testing, it was shown that there was no significant difference in OS [34]. While our data recapitulates that broad-based sequencing panels were not independently associated with better median OS, our results also show that patients with actionable alterations had a discernibly improved median OS as compared to wild-type patients who did not have an actionable alteration. Most notably ALK rearrangement patients had an improved OS as compared to tested ALK rearrangement negative (median OS 82.6 vs 26.6 months), which is suspected due to appropriate molecular testing at presentation and apt selection of therapy based on results [35, 36]. While Presley et al. include ALK in their small routine panel it should be noted that in a study evaluating the methods of ALK rearrangement testing it was found that comprehensive

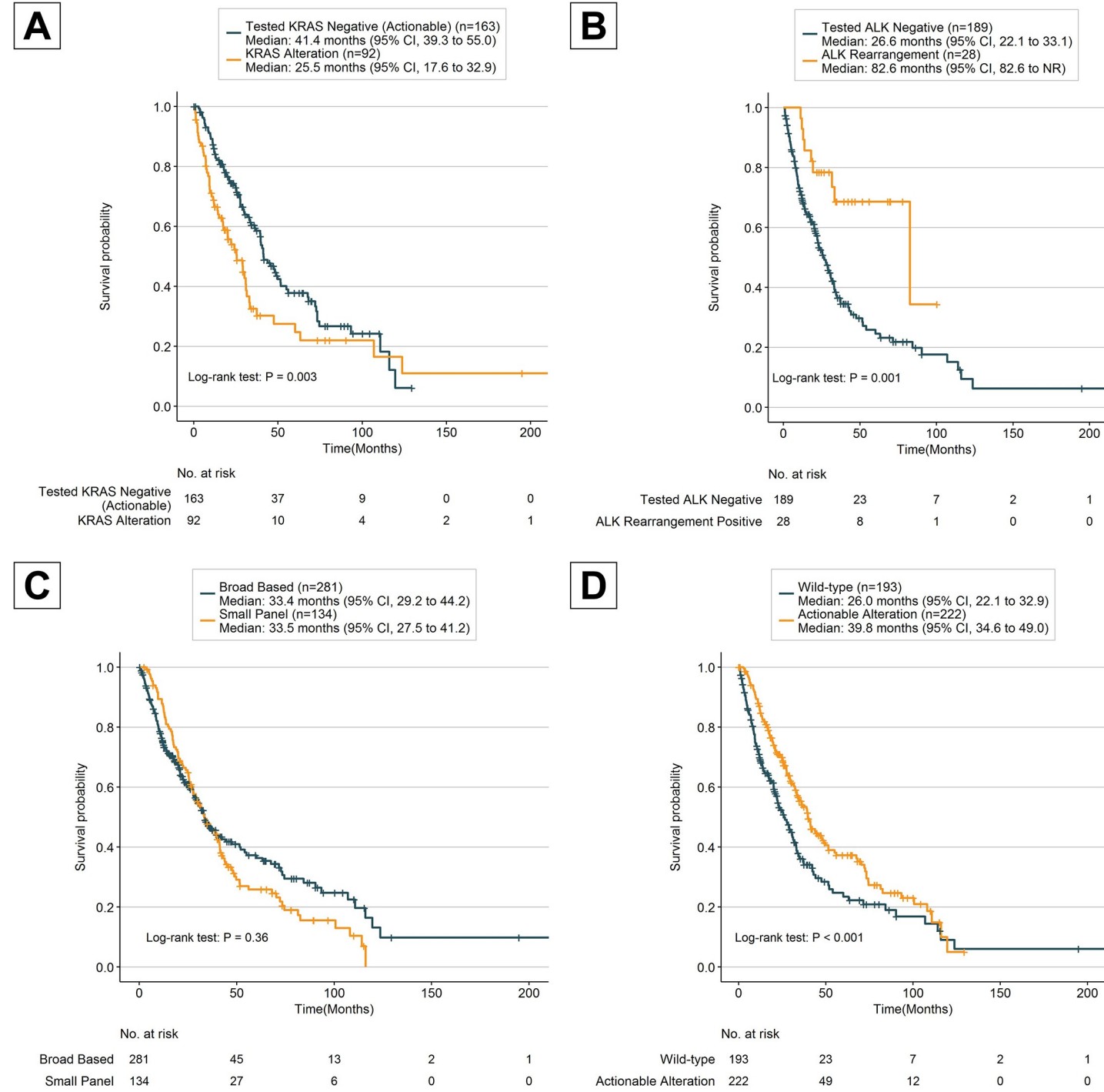

**Fig 2. Kaplan-Meier estimates of overall survival.** A) Overall survival among KRAS altered and tested KRAS negative patients with actionable alteration. B) Overall survival among ALK rearrangement and tested ALK rearrangement negative patients. C) Overall survival among broad-based panel tested and small-panel tested patients. D) Overall survival among actionable altered and wild-type patients. *P* values comparing risk groups were calculated with the log-rank test.

genomic profiling by NGS was able to identify a number of patients who had previously tested negative using the standard ALK fluorescence in situ hybridization (FISH) assay [37]. While there were several studies that showed improved PFS with targeted therapy which did not

translate into OS benefit [7–12], more recent results in lung cancer have shown there is a durable survival benefit for patients who are properly treated with appropriate inhibitors as opposed to patients who do not receive targeted therapy [30, 38–41]. 53.5% (222/415) of patients presented with actionable alterations in our cohort had specific targeted therapies approved by the FDA. While many question the utility to targeted therapy due to low actionable alteration rates [42–44], this study shows that actionable alterations rates may vary demographically and by location presenting a challenge for improving outcomes that may be addressed through the implementation of clinical pathway guidelines and continuing genomic education [45].

## Guided value-based medicine

Several studies have been performed that have determined the cost-effectiveness of NGS panels over single gene testing and in one study there was an improvement in PFS without additional health care costs to the patient [46–48]. In several meta-analyses of prospective clinical trials for solid tumors and hematological malignancies, personalized treatment strategies showed superior outcomes to those in control arms [49–54]. Therefore, the question becomes not whether genomic testing should be performed with a specific panel but whether the patient is treated accordingly based on all information available to improve their individual outcomes [54]. For this purpose, several associations including the National Comprehensive Cancer Network (NCCN) have issued guidelines which strongly emphasize not only testing for EGFR, ALK, ROS1, BRAF, and PD-L1 alterations but more importantly advise to conduct broader molecular profiling to identify rare mutational drivers [5, 6]. In practice, it was reported that the majority of oncologists (60%) in North America did not utilize genomic alteration results in their treatment decision making [55, 56].

To improve these statistics, the American Society of Clinical Oncology (ASCO) established in 2015 a task force on clinical pathways to improve treatment decision making through evidence-based clinical pathway guidelines [3]. Since then, ASCO revealed in a 2017 State of Cancer Care in America report a 42% increase from 2014 to 2016 of practices complying with a pathway program [57, 58]. Moreover, a recent study that evaluated 7 cancer programs, including COH (both academic and community), that employed clinical pathways demonstrated unprecedently high rates of molecular testing and concordant appropriate first-line treatment decision making based on actionable biomarkers [59]. Thusly, the end-goal for improvement of outcomes can only be achieved under the right circumstances where the appropriate testing is performed and the accurate treatment decisions are made.

## THOR in comparison with TCGA and GENIE

While there is some overlap in the alteration rates between our cohort and the publicly available datasets (TCGA/GENIE), there are more differences in the comparable rates of occurrences. Our study identified a disproportionally high rate of EGFR alterations with 49.9% (207/415) and actionable alteration rate with 53.5%, which was notably higher than the TCGA/GENIE rates [17, 18]. However, these results are consistent with a large gene profiling study of NSCLC from the Icahn School of Medicine at Mount Sinai that was able to identify actionable alterations in 65% of the cases [60]. While it is estimated that the actionability rate for NSCLC is around 30%, the results of this single-site analysis warrant further consideration to understand the heterogeneity of lung cancer in different populations and how this may impact survival outcomes [17]. This may in part be due to referrals to COH for clinical care and potential clinical trials, but it cannot be discounted that the high rates of EGFR alterations may be due to a large number of Asian never-smokers at presentation. In fact, 72.8% (99/136)

of Asians in our cohort presented with an EGFR alteration as compared with 50.0% EGFR incidence rate found in the GENIE dataset [17]. This large occurrence of EGFR in Asian never-smokers is consistent with another study that identified 75.3% of Asian never-smokers who harbored EGFR mutations [61].

Unexpectedly a number of chromatin modulating genes were irregularly prevalent in our cohort as compared with TCGA/GENIE. ARID1A alterations were highly expressed in our cohort alongside ARID1B and the two are known to be mutually exclusive within individual SWItch/Sucrose Non-Fermentable (SWI/SNF) chromatin remodeling complexes [62]. However, ARID1A has been shown to be the strongest tumor suppressor gene amongst the ARID genes in various cancers [63–66] and ARID1A is known to have the highest alteration rate among the SWI/SNF subunit genes [67]. The dysfunction of these complexes is known to destabilize the lung cancer genome by affecting chromatin remodeling and also disrupt DNA repair [68–70]. Furthermore, the high prevalence of genes such as MLL, MLL2, and BRCA2 suggest that other chromatin remodeling motifs may be involved in tumorigenesis of these patients rather than an event specific to a particular tumor type [71, 72].

## Conclusion

The limitation of current oncology practice is the speed of integration and adaptation to the rapidly advancing testing modalities and multiple therapeutic options that are garnering swift approval in various cancer types [73]. These efforts will require standardization of molecular testing modalities, cohesive guidelines, and pathways for precision medicine, and focused patient registries to enable cohort management and efficient clinical trial accrual. While our study was limited by the sample size and the demographics of our patients, it is still important to utilize these tools to guide physicians in determining which patient groups can benefit from clinical trials, tumor resistance or chemotherapy sensitivity. Review of the genomic data of our lung cancer patient cohorts in THOR showed significantly different incidences of actionable mutations as compared to the public datasets. Thus, instead of relying on generalized demographics and genomic results within the public datasets, the individual centers are required to perform their own assessments of their cohorts in order to have a proper analysis of their clinic population.

## Supporting information

**S1 Table. NGS platforms of testing.**
(DOCX)

**S2 Table. Actionable genomic alteration rates in TCGA and GENIE.**
(DOCX)

**S3 Table. Multivariate Cox Proportional Hazards Regression Models for Adjusted for Sex, Age, and Smoking Status.**
(DOCX)

**S4 Table. Cox Proportional Hazards Regression Models for Actionable Genes Adjusted for Sex, Age and Smoking Status.**
(DOCX)

**S1 File. Patient clinical and molecular marker data.** For molecular markers, "0" means tested negative, and blank means not tested.
(XLSX)

## Acknowledgments

We thank the clinical staff at City of Hope for their skill and dedication in helping the patients presented in this manuscript. The authors would like to acknowledge the American Association for Cancer Research and its support in the development of the AACR Project GENIE registry, as well as members of the consortium for their commitment to data sharing. Interpretations are the responsibility of study authors. The results published here are in whole or part based upon data generated by the TCGA Research Network: https://www.cancer.gov/tcga.

## Author Contributions

**Conceptualization:** Isa Mambetsariev, Yingyu Wang, Sorena Nadaf, Rebecca Pharaon, Idoroenyi Amanam, Arya Amini, Andrea Bild, Peiguo Chu, Loretta Erhunmwunsee, Jae Kim, Janet Munu, Raju Pillai, Dan Raz, Sagus Sampath, Lalit Vora, Erminia Massarelli, Marianna Koczywas, Karen Reckamp, Ravi Salgia.

**Data curation:** Isa Mambetsariev, Yingyu Wang, Chen Chen, Rebecca Pharaon, Jeremy Fricke, Idoroenyi Amanam, Fang Qiu.

**Formal analysis:** Isa Mambetsariev, Yingyu Wang, Chen Chen, Fang Qiu, Lynette Smith, Surinder K. Batra.

**Funding acquisition:** Ravi Salgia.

**Investigation:** Chen Chen, Rebecca Pharaon, Ravi Salgia.

**Methodology:** Isa Mambetsariev, Yingyu Wang, Chen Chen, Ravi Salgia.

**Project administration:** Sorena Nadaf, Erminia Massarelli, Karen Reckamp, Ravi Salgia.

**Resources:** Sorena Nadaf, Ravi Salgia.

**Supervision:** Sorena Nadaf, Erminia Massarelli, Karen Reckamp, Ravi Salgia.

**Validation:** Isa Mambetsariev, Chen Chen, Fang Qiu, Lynette Smith, Surinder K. Batra, Erminia Massarelli, Karen Reckamp.

**Visualization:** Isa Mambetsariev, Yingyu Wang, Chen Chen, Rebecca Pharaon.

**Writing – original draft:** Isa Mambetsariev, Yingyu Wang, Chen Chen, Sorena Nadaf, Rebecca Pharaon, Jeremy Fricke, Erminia Massarelli, Karen Reckamp, Ravi Salgia.

**Writing – review & editing:** Isa Mambetsariev, Yingyu Wang, Chen Chen, Sorena Nadaf, Rebecca Pharaon, Jeremy Fricke, Idoroenyi Amanam, Arya Amini, Andrea Bild, Peiguo Chu, Loretta Erhunmwunsee, Jae Kim, Janet Munu, Raju Pillai, Dan Raz, Sagus Sampath, Lalit Vora, Fang Qiu, Lynette Smith, Surinder K. Batra, Erminia Massarelli, Marianna Koczywas, Karen Reckamp, Ravi Salgia.

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
