## [Decision Letter · Decision Letter 0]

17 Oct 2019

PONE-D-19-20200

Precision Medicine and Actionable Alterations in Lung Cancer

PLOS ONE

Dear Dr. Ravi Salgia,

Thank you for submitting your manuscript to PLOS ONE. After careful consideration, we feel that it has merit but does not fully meet PLOS ONE’s publication criteria as it currently stands. Therefore, we invite you to submit a revised version of the manuscript that addresses the points raised during the review process.

We would appreciate receiving your revised manuscript by 31st October 2019. To enhance the reproducibility of your results, we recommend that if applicable you deposit your laboratory protocols in protocols.io, where a protocol can be assigned its own identifier (DOI) such that it can be cited independently in the future. For instructions see: http://journals.plos.org/plosone/s/submission-guidelines#loc-laboratory-protocols

We look forward to receiving your revised manuscript.

Kind regards,

Ramon Andrade De Mello, MD, FACP, PhD

Academic Editor

PLOS ONE

Journal Requirements:

2.  Please note that according to our submission guidelines (http://journals.plos.org/plosone/s/submission-guidelines), outmoded terms and potentially stigmatizing labels should be changed to more current, acceptable terminology. For example: “Caucasian” should be changed to “white” or “of [Western] European descent” (as appropriate).

3. As your study involved humans categorized by race/ethnicity, please:

- Explicitly describe the methods of categorizing human populations

- Define categories in as much detail as the study protocol allows

- Justify the choices of definitions and categories, including for example whether any rules of human categorization were required by the funding agency

- Explain whether (and if so, how) confounding variables such as socioeconomic status, nutrition, environmental exposures, or similar factors were controlled for in the analysis

For more information, please see our author guidelines: https://journals.plos.org/plosone/s/submission-guidelines#loc-human-subjects-research

Reviewers' comments:

Reviewer's Responses to Questions

**Comments to the Author**

1. Is the manuscript technically sound, and do the data support the conclusions?

Reviewer #1: Yes

Reviewer #2: Yes

2. Has the statistical analysis been performed appropriately and rigorously? 

Reviewer #1: Yes

Reviewer #2: Yes

3. Have the authors made all data underlying the findings in their manuscript fully available?

Reviewer #1: Yes

Reviewer #2: Yes

4. Is the manuscript presented in an intelligible fashion and written in standard English?

Reviewer #1: Yes

Reviewer #2: Yes

5. Review Comments to the Author

Reviewer #1: The authors nicely summarized the single institution experience in generating patient registry for comprehensive tumor genomic profiling in lung adenocarcinoma and its impact of patient survival. They found high EGFR mutation rate was likely driven by the enriched Asian population and clinical trials at the academic institution. Thus it is informative to the research community.

There are a few minor concerns:

1. The Title seems to be very general and does not reflect the study refers to single institutional experience.

2. The patients were diagnosed between 2008 and 2016, during which period the diagnosis and treatment of lung adenocarcinomas had evolved, it would be interesting to analyze if there were any differences between the lung cancer cases diagnosed in early and later years.

Reviewer #2: Motivated by the use of genomic tests for cancer treatment as well as the need for organizations to structure and use data to develop a personalized patient's treatment, the authors presents the use of genetic test of 415 patients with lung adenocarcinoma obtained by the Thoracic Oncology Registry. In general, the authors compared patients with genetic alteration with those without alteration (wild type) using statistical analyzes of association (chi-square test) and survival methods. The results show that the most frequent genetic alterations in oncogenes were in EGFR, KRAS and ALK, while the most commonly occurring tumor suppressor genes consisted of TP53, LRP1B and STK11. In addition, the authors compared the study's genomic alterations with other databases. Regarding the results, significant differences were observed in overall survival of patients with actionable change compared to wild type, justifying the use of patient registries and genomic data in cancer treatment. Although the statistical analysis is good, there are differences in the observed frequency for EGFG in table 1 and 2 (177 vs 207) and in table 2 the sum of observed frequency of exon 18 mutation, exon 19 deletion, exon 20 insertion, exon 21 mutation genes it is not equals to 207 (the presented sum is equal to 210). Furthermore, the authors proposed the use of cox proportional hazards models, but did not present or check the validity of proportionality assumption. What was the criteria for justifying a good adjust of the model to data? Also, besides the p-value, the authors should present the qui squared test statistic in table 1. It would be a good idea to check these inconsistencies.

6. PLOS authors have the option to publish the peer review history of their article (what does this mean?). If published, this will include your full peer review and any attached files.

Reviewer #1: Yes: Tianhong Li

Reviewer #2: No

---

## [Author Response · Author response to Decision Letter 0]

19 Nov 2019

Reviewer #1: The authors nicely summarized the single institution experience in generating patient registry for comprehensive tumor genomic profiling in lung adenocarcinoma and its impact of patient survival. They found high EGFR mutation rate was likely driven by the enriched Asian population and clinical trials at the academic institution. Thus it is informative to the research community.

There are a few minor concerns:

1. The Title seems to be very general and does not reflect the study refers to single institutional experience.

We thank the reviewer for their comment and have revised the title to “Precision Medicine and Actionable Alterations in Lung Cancer: A Single Institution Experience”.

2. The patients were diagnosed between 2008 and 2016, during which period the diagnosis and treatment of lung adenocarcinomas had evolved, it would be interesting to analyze if there were any differences between the lung cancer cases diagnosed in early and later years.

In our evaluation of patients who were diagnosed in the early years, we found that the majority of patients only had molecular testing on a small panel of genes, including EGFR, ALK, and KRAS. With the advent of NGS, patients in the later years had larger panels of genes and alterations that were identified, including ROS1 fusion, MET exon 14 alterations and others such as TP53 and STK11. However, as shown in our results, the difference in panels did not show a significant difference in overall survival. Nevertheless, there was a difference in survival between patients with actionable alterations versus those without actionable alterations. As mentioned in our discussion, more and more genes and alterations are receiving FDA approval for targeted therapy and it will be important to continue to screen patients for all actionable alterations to improve their outcomes. Most recently several MET exon 14 deletion targeted therapies, such as tepotinib and capmatinib, have received FDA breakthrough therapy designation with significant improvements in patient outcomes. 

Reviewer #2: Motivated by the use of genomic tests for cancer treatment as well as the need for organizations to structure and use data to develop a personalized patient's treatment, the authors presents the use of genetic test of 415 patients with lung adenocarcinoma obtained by the Thoracic Oncology Registry. In general, the authors compared patients with genetic alteration with those without alteration (wild type) using statistical analyzes of association (chi-square test) and survival methods. The results show that the most frequent genetic alterations in oncogenes were in EGFR, KRAS and ALK, while the most commonly occurring tumor suppressor genes consisted of TP53, LRP1B and STK11. In addition, the authors compared the study's genomic alterations with other databases. Regarding the results, significant differences were observed in overall survival of patients with actionable change compared to wild type, justifying the use of patient registries and genomic data in cancer treatment. Although the statistical analysis is good, there are differences in the observed frequency for EGFG in table 1 and 2 (177 vs 207) and in table 2 the sum of observed frequency of exon 18 mutation, exon 19 deletion, exon 20 insertion, exon 21 mutation genes it is not equals to 207 (the presented sum is equal to 210). Furthermore, the authors proposed the use of cox proportional hazards models, but did not present or check the validity of proportionality assumption. What was the criteria for justifying a good adjust of the model to data? Also, besides the p-value, the authors should present the qui squared test statistic in table 1. It would be a good idea to check these inconsistencies.

We thank the reviewer for their detailed comments and have addressed their concerns below.

Regarding the discrepancy in EGFR incidence between Table 1 and Table 2: In Table 1 the EGFR is labeled “EGFR (L858R/exon 19 deletion)” and therefore only includes EGFR patients who had those two alterations. The reason for this is because they are the only two types of EGFR alterations that are considered “actionable” by the FDA and eligible for EGFR TKIs. Meanwhile, Table 2 includes all types of EGFR alterations including exon 18 mutation, exon 20 insertion and others. Hence, why there is a discrepancy of 177 vs 207. The methods for testing the assumption of proportional hazards was provided in the methods of the manuscript. Furthermore, the Cox models were adjusted using time-dependent variables to adjust the non-proportionality of variables. The revised Cox models are presented in Table 4, Supplementary Table 3, and Supplementary Table 4. Table 1 was revised to include the chi-squared test statistics.

---

## [Editor Report · Decision Letter 1]

10 Jan 2020

Precision medicine and actionable alterations in lung cancer: a single institution experience

PONE-D-19-20200R1

Dear Dr. Ravi Salgia,

We are pleased to inform you that your manuscript has been judged scientifically suitable for publication and will be formally accepted for publication once it complies with all outstanding technical requirements.

With kind regards,

Ramon Andrade De Mello, MD, PhD, FACP

Academic Editor

PLOS ONE

---

## [Editor Report · Acceptance letter]

3 Feb 2020

PONE-D-19-20200R1 

Precision medicine and actionable alterations in lung cancer: a single institution experience 

Dear Dr. Salgia:

I am pleased to inform you that your manuscript has been deemed suitable for publication in PLOS ONE. Congratulations! Your manuscript is now with our production department. 

With kind regards,

on behalf of

Professor Ramon Andrade De Mello 

Academic Editor

PLOS ONE